# Optimization of Hydrogen Yield from the Anaerobic Digestion of Crude Glycerol and Swine Manure

**Aguilar-Aguilar F. A. [1],\* , Adriana Longoria [1] , Juantorena A. U. [2] and P. J. Sebastian [1],\***

[1] Laboratorio de Bioenergía, Instituto de Energías Renovables (IER-UNAM), Temixco Morelos 62580, México; adlon@ier.unam.mx

[2] Facultad de Ciencias Agropecuarias, Universidad Autónoma del Estado de Morelos (UAEM), Cuernavaca Morelos 62210, México; alinajt2@yahoo.es

\* Correspondence: faguilar@ier.unam.mx (A.-A.F.A.); sjp@ier.unam.mx (P.J.S.); Tel.: +52-961-659-9559 (A.-A.F.A.); +52-777-362-0090 (ext. 29752) (P.J.S.)

**Abstract:** Crude glycerol and swine manure are residues with exponential production in Mexico, nonetheless, they have the potential to generate hydrogen from the fermentation process. For this reason, this study has evaluated the optimization of hydrogen yield from crude glycerol and swine manure, using the response surface methodology. The response surface methodology helps in the compression of the mixture of crude glycerol/ swine manure, with the production of hydrogen as a result, which improves the yields of the process, reducing variability and time of development. A central composite design was employed with two factors, six axial points and four central points. The two factors evaluated were crude glycerol and swine manure concentrations, which were examined over a range of 4 to 10 g L$^{-1}$ and 5 to 15 g L$^{-1}$, respectively. This study demonstrated that the thermal pretreatment method is still the most suitable method to be applied, mainly in the preparation of hydrogen-producing inoculum. The maximum hydrogen yield was 142.46 mL per gram of volatile solid added. It used up 21.56% of the crude glycerol (2.75 g L$^{-1}$) and 78.44% (10 g L$^{-1}$) of the swine manure, maintaining a carbon/nitrogen ratio of 18.06, with a fermentation time of 21 days. The response surface methodology was employed to maximize the hydrogen production of crude glycerol/swine manure ratios by the optimization of factors with few assays and less operational cost.

**Keywords:** hydrogen; anaerobic digestion; crude glycerol; swine manure

## 1. Introduction

Hydrogen is a promising candidate as a clean and renewable fuel, since it generates high-performance energy (122 kJ g$^{-1}$) and during its combustion, it generates water instead of polluting gases such as $CO_2$ and $NO_x$ [1,2]. This fuel is an alternative to fossil fuels, since it can be obtained from renewable and low-cost sources. However, hydrogen is not freely available in the nature; therefore currently, the biggest challenge is finding an efficient way to produce it from renewable sources at a low cost [3,4]. At present, 40% of hydrogen is generated from the catalytic oxidation of natural gases, 30% from heavy metals and naphtha, 18% from coal, 4% by electrolysis, and about 1% by biological processes. Obtaining hydrogen from biological processes is of interest, since it can potentially be an inexhaustible form of production [5].

Studies on hydrogen production by biological processes have focused mainly on direct or indirect bio-photolysis using algae and cyanobacteria, photo-fermentation of the organic compounds by photosynthetic bacteria, and dark fermentation of organic compounds with anaerobic bacteria. However, bio-photolysis and photo-fermentation have disadvantages, since the hydrogen-producing

enzymes are very sensitive to the presence of oxygen, the need for constant light, and low yields of hydrogen [6,7].

The generation of hydrogen by dark fermentation could be the most favorable, since it can be generated continuously with high production rates, compared to other biological processes [8,9]. Additionally, if organic waste is used, it is feasible to obtain a product with added value, by the treatment of waste that is generated in the agroindustry. Angelidaki 1994 [10] evaluated the effect of temperature in the range of 40–64 °C, of cattle manure with two different ammonia concentrations (2.5 and 6.0 g $L^{-1}$). They demonstrated that a free ammonia concentration of over 0.7 g $L^{-1}$ and a temperature of above 55 °C had negative effects in reactors, with an increase in volatile fatty acids. In a study by Kotsopoulos et al. 2009 [8], pig slurry was used to produce hydrogen under thermophilic temperatures (70 °C). During the fermentation period, the hydrogen yield was 3.65 mL per gram of volatile solid added. The high temperature of the reactor enhanced hydrogen production in pig slurry; nevertheless, volatile fatty acid (VFA) production also increased, obtaining a low hydrogen yield.

In order to avoid inhibition by nitrogen in the fermentation process, the study of Zhu et al., 2009 [11] used pig slurry in anaerobic co-digestion with glucose to produce hydrogen; the yield was 209 mL $g^{-1}$ of volatile solid added; they also used pH control to keep the pH at 5.3 in the system. Therefore, the process is unviable, since they used glucose as a substrate and a chemical agent to regulate the pH. Likewise. Salerno et al., 2006 [12] studied the production of hydrogen from glucose with different concentrations of ammonium. They observed that the highest hydrogen yield was 170 mL, with 0.8 g $L^{-1}$ ammonium concentration, though 7.8 g $L^{-1}$ of ammonium concentration decreased hydrogen production to 105 mL.

On the other hand, residues with a high concentration of carbohydrates tend to increase the concentration of volatile fatty acids in the fermentative medium. Crude glycerol, which comes from the production of biodiesel, is a high source of carbohydrates as energy for anaerobic microorganisms, which could be used to produce hydrogen. However, the presence of ethanol, fatty acid esters, mono-, di-, and triglycerides, and methyl esters affect microbial metabolism [13]. Mangayil et al., 2012 [14] studied the production of hydrogen with different concentrations of crude glycerol. In the study, they observed that 1 g $L^{-1}$ of crude glycerol has a maximum yield of 16.1 mL of hydrogen, while 5 g $L^{-1}$ generated only 5.5 mL. They concluded that the crude glycerol impurities create inhibitory effects in the fermentation process; therefore, a complementary co-substrate is necessary to increase the yield of hydrogen.

These observations suggest the potential for the co-digestion of swine manure and residues with high carbohydrate contents, to produce hydrogen by dark fermentation successfully. In this context, the co-digestion process must even be contemplated locally within the same agro-industrial system, to directly use the waste, in order to optimize the rate of the load of the C/N (carbon/nitrogen) ratio by dilution, to avoid other inhibitors. This should, consequently increase the stability of the biological process in the fermentative process to produce hydrogen. As observed in the study by Li et al., 2008 [15], the maximum hydrogen yield was 193.85 mL $g^{-1}$ volatile solids from fruit and vegetable residues, and aged refuse excavated from a typical sanitary landfill (100:50). It is probably the balance in the C/N ratio and the synergy between the residues that improved the process of producing hydrogen by dark fermentation. Therefore, different complementary and less expensive forms of organic waste must be evaluated as supplementary nutrients to produce hydrogen.

The response surface methodology offers advantages in comparison to the conventional statistical methods. It helps with the understanding of data for swine manure (SM) co-digested with crude glycerol (CG) (CG/SM ratios) and the responses, increases the hydrogen yields in fermentation, reduces the time of development, and lowers costs. Biological processes generally use common statistical methods; however, RSM is an established method that has been used, as in our already-reported work [9], and in other bioprocess studies [16].

The aim of this research was to explore the viability of SM co-digested with CG, in batch experiments of hydrogen production under mesophilic conditions. The residues used in this study

were waste from the biodiesel production process. Pig farming plays an important role in the economy of Mexico, through the production of meat and its derivatives. Generally, the crude glycerol from biodiesel production and swine manure are residues without destination for disposal, with high costs and environmental impacts.

## 2. Results

### 2.1. Characterization of the Substrates

In the anaerobic digestion process, it is important to know the physicochemical composition of the substrates, which play an important role for maximizing the hydrogen yield and reducing the volume of the waste. From the physicochemical characterization of CG and SM (Table 1), the VS/TS (volatile solid/total solid) ratio was established, which represents the percentage of the total organic matter of each residue; SM contained 83.83% organic matter and CG had almost 100% biodegradability. Labatut et al., 2011 [17] demonstrated that the VS/TS ratio, an indirect measure, correlates with their biodegradability by microorganisms during anaerobic digestion process. Accordingly, the CG and SM used in our study show potential for being employed as substrates for the anaerobic digestion process. In the same way, the C/N ratio is an important parameter that is related to the required nutrient levels in a substrate for anaerobic microorganisms. An elevated C/N ratio induces a low rate of protein solubility and leads to low concentrations of free ammonia in the reactor [18]. As reported by Wang et al., 2014 [19], the largest C/N ratios studied in anaerobic digestion have been in the range of 10 to 35 carbons to 1 (one) nitrogen. Therefore, insufficient amounts of carbon or nitrogen may limit the anaerobic fermentation performance, when CG and SM are used independently [13,20].

**Table 1.** Physical and chemical characteristics of hydrogen production assays conducted with different concentrations of substrates, crude glycerol (CG), and swine manure (SM).

| Parameters | CG | SM | Inoculum |
|:---:|:---:|:---:|:---:|
| pH | $10.35 \pm 0.00$ | $6.513 \pm 0.53$ | $6.73 \pm 0.15$ |
| TS (g L$^{-1}$) | $870.34 \pm 0.00$ | $199.86 \pm 0.01$ | $6.77 \pm 0.60$ |
| VS (g L$^{-1}$) | $870.09 \pm 0.00$ | $167.55 \pm 0.01$ | $5.04 \pm 0.69$ |
| VS/TS (% m m$^{-1}$) | 99.97 | 83.83 | 74.39 |
| COD (g L$^{-1}$) | 1974.40 | 137.83 | 5.47 |
| BOD (g L$^{-1}$) | 1934.91 | 83.41 | n/d |
| BOD/COD | 0.98 | 0.61 | n/d |
| Carbon (% w w$^{-1}$) | 88.04 | 49.62 | n/d |
| Nitrogen (% w w$^{-1}$) | <0.05 | 4.08 | n/d |
| Oxygen (% w w$^{-1}$) | 11.08 | 6.50 | n/d |
| Protein (% w w$^{-1}$) | <0.05 | 25.55 | n/d |
| C/N Ratio | 1760.80 | 12.14 | n/d |

n/d: not determined, TS: Total Solids, VS: Volatile Solids, COD: chemical oxygen demand, BOD: biochemical oxygen demand, C/N (carbon/nitrogen) ratio.

According to Dennehy et al., 2016 [21], the addition of complementary nutritional components to the digestion of residues is a widely applied procedure in order increase the hydrogen production, since such components can provide nutrient balance, C/N (carbon/nitrogen) ratio balance, and reduced costs related to pH control, by providing a buffering capacity in the reactor, which is necessary for the optimization of the hydrogen production process.

### 2.2. Inoculum Selection with Thermal and Acid Pretreatments in Hydrogen Production

#### 2.2.1. Kinetics of Hydrogen Yield

Hydrogen production from inoculation with acid and thermal pretreatments is shown in Figure 1, where significant differences were observed in the hydrogen yield as function of time. Besides,

all of the assays gave a monophasic curve of hydrogen production, and the exponential production rate was between 10–21 days of incubation. It was also observed that the adaptation stages of the microorganisms were different for each treatment. For example, for the assays inoculated with thermal pretreatments, the adaptation of the microorganisms required eight days, a longer period in comparison with the tests that were inoculated with acid pretreatment, which required approximately two days. It is probable that the assays inoculated with thermal pretreatment (120 °C, 60 min and 1 atm) inhibited the growth of most methanogenic bacteria, with longer adaptation times as a response. Furthermore, the conversion of the substrate was slow, and the hydrogen production was higher compared to that under acid enrichment, as substrate consumption started more quickly, and finished in less time.

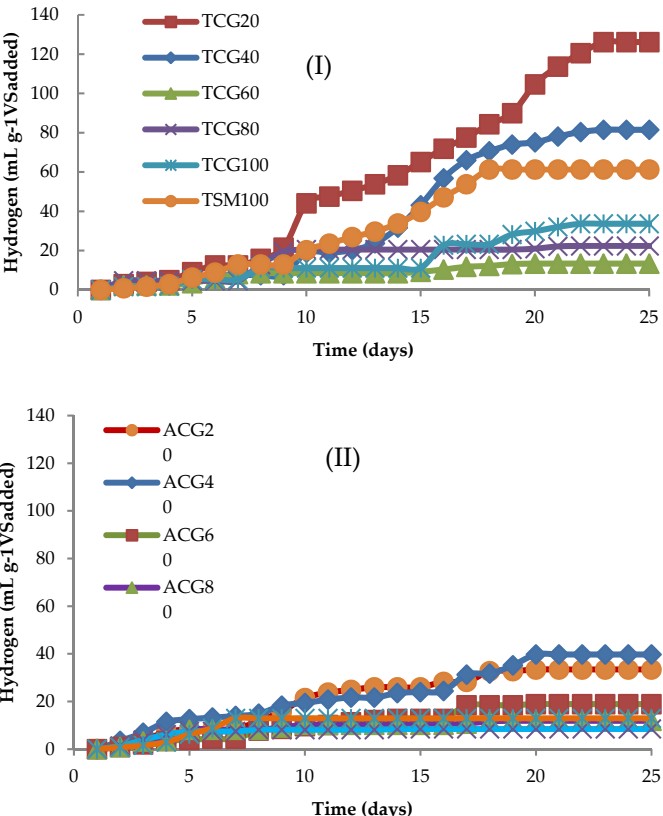

**Figure 1.** Comparison of the kinetics of hydrogen production with (**I**) thermal pretreated inoculum (T) and (**II**) acid pretreated inoculum (A). In assays were added different concentrations of substrates: CG20 (20% CG:80% SM), CG40 (40% CG:60% SM), CG60 (60% CG:40% SM), CG 80 (80% CG:20% SM), CG100 (100% CG), SM100 (100% SM).

Higher hydrogen production values with the inoculant enriched with thermal pretreatment were observed in TCG20, TCG40, and TSM100, with 126.14 mL gVS$^{-1}$, 81.51 mL gVS$^{-1}$, and 61.25 mL gVS$^{-1}$, respectively. TCG60, TCG80, TCG100, and assays with inoculum enriched with acid pretreatment generated lower hydrogen concentrations (<40 mL gVS$^{-1}$) (Table 2). It was observed that the rapid hydrogen production from using acid-enriched inoculum decreased with time of adaptation of the microorganisms but increased the concentration of volatile fatty acids (VFAs) in the reactor, and the pH also decreased (between 5–6). It is also possible that increasing the concentration of the carbon source (CG) accelerates the fermentation process and inhibits the hydrogen-producing bacteria. This study demonstrates that the thermal pretreatment method is still the most suitable method that has been applied, mainly in the preparation of hydrogen-producing inoculants. The results of this study were consistent with those authors who made selections of hydrogen-producing bacteria [22–26]. Wang and Wan 2008 [27] compared five pretreatment methods (acid, base, thermal, aeration, and chloroform) for the selection of hydrogen-producing bacteria, and successfully inhibited

the growth of methanogenic bacteria, including hydrogenotrophic and acetoclastic methanogenics. They demonstrated that for fermentative hydrogen production from glucose, an inoculum that was pretreated by thermal enrichment could obtain the maximal hydrogen yield potential. As reported by Rafieenia et al. 2018 [28], the assays inoculated with thermal enrichment inoculum were used as the main inoculum enrichment method in hydrogen fermentation. The main hydrogen producers present in anaerobic mixed microflora after pretreatment were from the genus *Clostridium spp.* and *Bacillus spp.* spore forming microorganisms. Methanogens that are the main hydrogen consumers are very sensitive to certain temperatures while spore-forming hydrogen producers can resist high temperatures by sporulation.

**Table 2.** Production of hydrogen with different concentrations of crude glycerol (CG) and swine manure (SM) with thermal- and acid-pretreated inocula.

| SM(%) | CG(%) | C/N Ratio | $pH_{initial}$ | Hydrogen (mL $g^{-1}VS_{added}$) | |
|---|---|---|---|---|---|
| | | | | Thermal Pretreatment | Acid Pretreatment |
| 80 | 20 | 16.24 | 7.38 ± 0.08 | [c] 126.14 ± 12.3 | [a] 33.44 ± 10.1 |
| 60 | 40 | 23.04 | 7.29 ± 0.20 | [d] 81.51 ± 9.9 | [a] 39.75 ± 4.9 |
| 40 | 60 | 36.48 | 7.33 ± 0.15 | [e] 13.38 ± 7.4 | [b] 18.92 ± 5.6 |
| 20 | 80 | 75.57 | 7.50 ± 0.11 | [e] 22.39 ± 10.2 | [b] 11.89 ± 3.1 |
| 0 | 100 | 1760.80 | 7.26 ± 0.23 | [d] 33.70 ± 6.98 | [b] 8.46 ± 4.4 |
| 100 | 0 | 12.14 | 7.00 ± 0.04 | [d] 61.25 ± 15.4 | [b] 12.94 ± 2.7 |

Different symbols (a, b, c, d, e) represent significant differences according to a confidence level of 5%, in hydrogen production.

Another parameter that affects the hydrogen yield is the C/N ratio. In our study, the low concentration of the carbon source (CG) and the high concentration of nitrogen (SM) improved the hydrogen yield, as was demonstrated by the TCG20 assay (20% CG/80% SM), with a C/N ratio of 16.24 (Figure 1 and Table 2); it maximized the hydrogen yield of 126.14 mL $g^{-1}$ of VS added. In contrast, the assay CG80 (80% CG/20% SM) showed a low hydrogen yield of 22.39 mL, using thermal enriched inoculum and 11.89 mL $g^{-1}$ of VS added, using an acid-enriched inoculum, with a C/N ratio of 75.57. This study demonstrated that the high concentration of crude glycerol (>25% in weight) increased the C/N ratio and decreased the hydrogen yield.

### 2.2.2. Effect of CG/SM Ratios on Reactors with Thermal and Acid Inocula

The initial pH is another key factor for the improvement of hydrogen production. Some authors suggest an optimal initial pH that is between 5.5 and 6.0 [29,30]. However, in this study, the initial pH in the reactors inoculated with thermal- and acid-enriched inocula was above 7 in all cases (Table 2). Specifically, the highest and lowest hydrogen yields were obtained at an initial pH of 7.38 and 7.50, respectively, and the final pH of all reactors was 4–6. Likewise, the pH was between the values reported by Dong et al., 2010 [30]; in their study, they reported that the final pH in the batch reactor was approximately between 4.6 and 5.3, using the organic fraction of urban solid waste as a substrate. Ren et al., 2017 [31] mentioned that the optimum pH for the development of methanogenic bacteria is between 6.5–7.5, outside of this range, the methane production inhibition is guaranteed. Therefore, the hydrogen producing bacteria are favored in fermentation process.

In general, when a single substrate is used in anaerobic digestion, very little hydrogen can be generated (8.46–33.70 mL $g^{-1}$ $VS_{added}$ for CG, and 61.25–12.94 mL $g^{-1}$ $VS_{added}$ for SM). However, the hydrogen produced with swine manure and other co-substrates can vary, depending on the biodegradability and complexity of the mixture of substrates. The hydrogen production study carried out by Tenca et al., 2011 [32] used fruit and vegetable residues with swine manure (35/65 ratio), obtaining a hydrogen yield of 165 mL $g^{-1}$ of VS. Nevertheless, in the study by Viana et al., 2012 [13], the direct use of CG is not recommended, firstly due to high COD concentrations, and secondly, due to

the high concentrations of $Cl^{-1}$ (34–46 g $L^{-1}$), from the HCl used for the neutralization of glycerol, which comes from the alkaline biodiesel production industry. Since the high concentration of chlorine ions inhibit bacterial growth and consequently affect hydrogen production. Interestingly, the thermal enrichment of the inoculum and the combination of the crude glycerol and swine manure improve anaerobic digestion; the presence of higher concentrations of SM (80%) increased the degradation efficiency of CG.

### 2.3. Increases in the Organic Load of the CG/SM Ratio with Thermal Inoculum

### 2.3.1. Characterization of the CG/SM Combination

The physicochemical characteristics of the mixtures of the substrates (CG and SM, Table 3) used in this study were in the range of 4.39 to 11.78 g $L^{-1}$ of TS; 4.08 to 11.45 g $L^{-1}$ of VS, and 6.12 to 19.41 g $L^{-1}$ of COD. Regarding the inoculum/substrate ratio, it was within the range mentioned by Labatut et al., 2011 [17]. They observed that the volatile solids of the manure in anaerobic digestion must be greater than 3 g $L^{-1}$, and a minimum I/S (inoculum/substrate) ratio of 0.5, to ensure the commencement of the process during the first 3 day of testing. In the same way, the C/N ratio affects the nutrient levels of the substrate in digestion, and thus, a high C/N ratio induces a low protein solubility rate, and it leads to low concentrations of free ammonia in the system. In accordance with Zhang et al., 2013 [33] and Elsayed et al., 2017 [34], the C/N ratio used in anaerobic digestion is between 10 and 35 carbons for one nitrogen, a ratio that is 25 times more than that which is commonly used. Insufficient amounts of carbon or nitrogen can limit the fermentation performance of substrates, when CG and SM are used independently.

**Table 3.** Physicochemical characterization of the mixture of crude glycerol (CG) and swine manure (SM) prior to the anaerobic digestion to produce hydrogen.

| Assays | Concentration of the Combined Substrates (g $L^{-1}$) | | $COD_l$ (g $L^{-1}$) | TS (g $L^{-1}$) | VS (g $L^{-1}$) | C/N Ratio (g $g^{-1}$) |
|---|---|---|---|---|---|---|
| | CG | SM | | | | |
| 1 | 4.00 | 5.00 | 7.17 ± 0.35 | 4.48 ± 0.13 | 4.32 ± 0.08 | 29.37 |
| 2 | 4.00 | 15.00 | 8.98 ± 0.79 | 6.48 ± 0.29 | 5.99 ± 0.13 | 17.88 |
| 3 | 10.00 | 5.00 | 16.56 ± 0.35 | 9.70 ± 0.39 | 9.54 ± 0.37 | 55.22 |
| 4 | 10.00 | 15.00 | 18.37 ± 0.26 | 11.70 ± 0.80 | 11.21 ± 0.67 | 26.50 |
| 5 | 2.75 | 10.00 | 6.12 ± 0.18 | 4.39 ± 0.37 | 4.07 ± 0.36 | 18.06 |
| 6 | 11.24 | 10.00 | 19.41 ± 0.35 | 11.78 ± 1.15 | 11.45 ± 1.07 | 36.35 |
| 7 | 7.00 | 2.93 | 11.49 ± 0.78 | 6.68 ± 0.70 | 6.58 ± 0.74 | 63.63 |
| 8 | 7.00 | 17.07 | 14.05 ± 0.10 | 9.50 ± 1.07 | 8.95 ± 0.97 | 20.97 |
| 9 | 7.00 | 10.00 | 12.77 ± 0.47 | 8.09 ± 0.27 | 7.77 ± 0.27 | 27.22 |
| 10 | 7.00 | 10.00 | 12.77 ± 0.47 | 8.09 ± 0.27 | 7.77 ± 0.27 | 27.22 |
| 11 | 7.00 | 10.00 | 12.77 ± 0.47 | 8.09 ± 0.27 | 7.77 ± 0.27 | 27.22 |
| 12 | 7.00 | 10.00 | 12.77 ± 0.47 | 8.09 ± 0.27 | 7.77 ± 0.27 | 27.22 |

CG: crude glycerol, SM: swine manure, COD: chemical oxygen demand, TS: total solid, VS: volatile solid.

### 2.3.2. Kinetics of the Hydrogen Yield for Different Concentrations of the CG/SM Ratio

The results obtained allowed to infer that the production of hydrogen by mixed cultures is a function of the fermentative metabolism of the bacteria, and the interactions with the substrates. It was observed that the different concentrations of substrates directly affected the overall performance of the fermentation process, as it was possible to observe the in assays 3, 4, and 6, with a high concentration of a mixture of substrates, having the lowest hydrogen yield with 69.86, 63.57, and 52.76 mL per gram of VS added (Table 4). Conversely, it was observed that in assay 5 with lower substrate concentrations (4.06 g VS) there was a higher efficiency in the production of hydrogen, with 142.46 mL during the 21-day fermentation (Figure 2). Marone et al., 2015 [35] used cheese whey residues and buffalo slurry,

reaching 117 mL of hydrogen per gram of VS added. In their study, the synergetic effect of the crude glycerol (33%) with buffalo slurry (66%) was observed, improving hydrogen production to 116 mL $g^{-1}$ of COD added; the initial pH was 6.5 and final pH was 4. However, the single digestion of crude glycerol and buffalo slurry affected the hydrogen yield (32 mL $g^{-1}$ of VS and 2.8 mL $g^{-1}$ of VS).

**Table 4.** Final parameters from the fermentative process, carried out on assays with different concentrations of crude glycerol and swine manure.

| Assays | Concentration of Combined Substrates (%) | | Hydrogen (mL $g^{-1}$ VS$_{added}$) | Accumulated Hydrogen (mL) |
|---|---|---|---|---|
| | CG | SM | | |
| 1 | 44.44 | 55.56 | 71.78 ± 5.90 | 7.75 ± 0.37 |
| 2 | 21.05 | 78.95 | 86.36 ± 6.68 | 12.94 ± 0.33 |
| 3 | 66.67 | 33.33 | 69.86 ± 9.88 | 16.66 ± 1.37 |
| 4 | 40.00 | 60.00 | 63.57 ± 2.85 | 17.82 ± 0.55 |
| 5 | 21.56 | 78.44 | 142.46 ± 3.24 | 14.49 ± 0.63 |
| 6 | 52.92 | 47.08 | 52.76 ± 4.84 | 15.11 ± 0.94 |
| 7 | 70.51 | 29.49 | 103.61 ± 6.71 | 17.05 ± 0.91 |
| 8 | 29.08 | 70.92 | 69.95 ± 4.97 | 15.65 ± 0.99 |
| 9 | 41.18 | 58.82 | 71.84 ± 4.69 | 13.95 ± 0.81 |
| 10 | 41.18 | 58.82 | 77.82 ± 5.33 | 15.11 ± 1.04 |
| 11 | 41.18 | 58.82 | 72.23 ± 5.05 | 14.02 ± 0.93 |
| 12 | 41.18 | 58.82 | 64.25 ± 5.57 | 13.89 ± 1.08 |

CG: crude glycerol, SM: swine manure, VS: volatile solid.

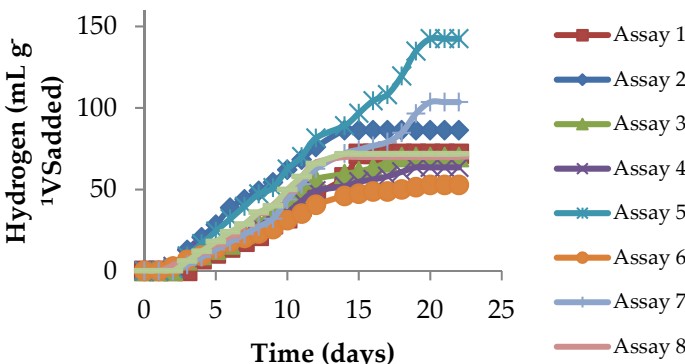

**Figure 2.** Profile of the hydrogen production tests with 21 day of fermentation of crude glycerol and swine manure. Assay 1 (4.31 g VS $L^{-1}$); Assay 2 (5.99 g VS $L^{-1}$); Assay 3 (9.53 g VS $L^{-1}$); Assay 4 (11.21 g VS $L^{-1}$); Assay 5 (4.07 g VS $L^{-1}$); Assay 6 (11.45 g VS $L^{-1}$); Assay 7 (6.58 g VS $L^{-1}$); Assay 8 (8.95 g VS $L^{-1}$); Assay 9 (7.76 g VS $L^{-1}$).

The adaptation phase of the microbial flora was for three days, the exponential phase of consumption of hydrogen production was in the period from day 3 to day 18, the cellular reproduction was extremely active and with greater metabolic activity the stationary phase was observed with a slow consumption of the substrates (Figure 2). In this phase, the microorganisms were sensitive to environmental changes in the system, probably due to the increase in the concentration of organic acids in the medium, with a fall in pH to less than 6, affecting cell development and hydrogen production. Fernandes et al. 2010 [36] generated hydrogen from crude glycerol with 200 mL $g^{-1}$ COD, domestic sewage with 200 mL $g^{-1}$ COD, vinasse with 579 mL $g^{-1}$ COD, and sucrose with 200 mL $g^{-1}$ COD. In comparison with our study, Fernandes et al., 2010 [36] regulated the pH with chemical agents, such as sodium bicarbonate, which can generate extra costs in the production of hydrogen. In our study, the buffer capacity was from the mixture of residues and the best results were with 21.56% of CG

and 78.44% of SM (Table 4). Our results demonstrate that the combination of the CG with 2.75 g L$^{-1}$ and 10 g L$^{-1}$ of SM generates hydrogen without external agents to regulate the pH.

### 2.3.3. Analysis of the Response Surface and the Analytical Model

The adjustment of the polynomial model generated to produce hydrogen as a function of different ratios of CG and SM yielded a determination coefficient (R2adj). Figure 3 shows the R$^2$ value obtained from the response surface methodology (RSM); the coefficient of determination (R$^2$) was 0.916, thus explaining the variability of 91.6% of the response variable. This fact was confirmed by obtaining a similar value for the adjusted determination coefficient (R$^2$adj = 0.864), which indicates that the model generated a positive response from the experimental data (hydrogen yield).

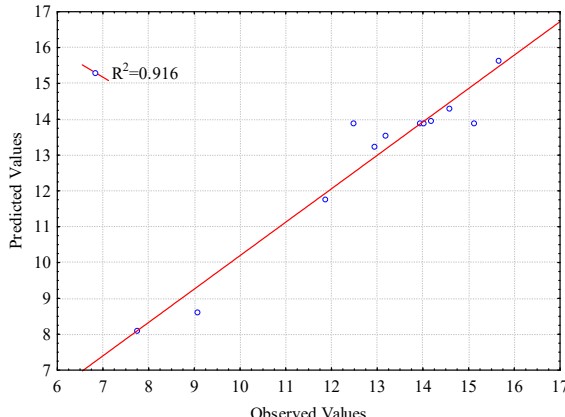

**Figure 3.** Prediction of R$^2$ in the experimental data and the data predicted by the model, using response surface methodology (RSM).

The variance analysis (ANOVA) resulted in a model of adjustment in the production of hydrogen that was highly significant, confirming that the model was able to adequately represent the data in the experimental region, showing a significant effect in the response, with a level of significance of 5%, as illustrated by the Pareto chart (Figure 4). According to hydrogen production statistical analysis, it can be affirmed that the linear terms of SM and CG were significant, according to *p*-value = 0.003 and 0.01, respectively, contrary to the quadratic term of SM that was not significant (*p*-value > 0.63). In addition, it was observed that the value of the quadratic effect of the CG factor was positive, indicating that its increase influences the increase in the volume of hydrogen.

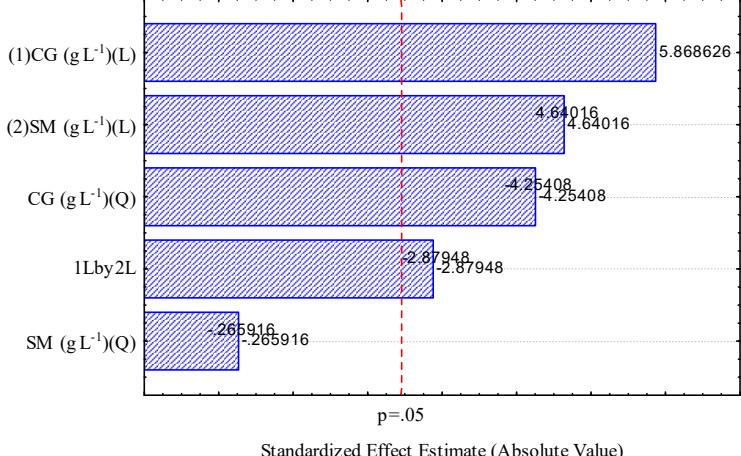

**Figure 4.** Pareto chart for applying the response surface methodology (RSM) model in the hydrogen production of crude glycerol (CG) and swine manure (SM).

In the analysis of the response surface graph generated from Equation (1) (Figure 5A) and Equation (2) (Figure 5B), it was possible to observe the positive effects of the CG and SM factors on hydrogen production.

$$Hydrogen\left(mL\ g^{-1}VS\right) = 157.19 - 15.63X - 0.79Y + 0.91X^2 - 0.35XY + 0.11Y^2 \tag{1}$$

$$Accumulated\ hydrogen = 9.17 + 0.61X + 0.11Y \tag{2}$$

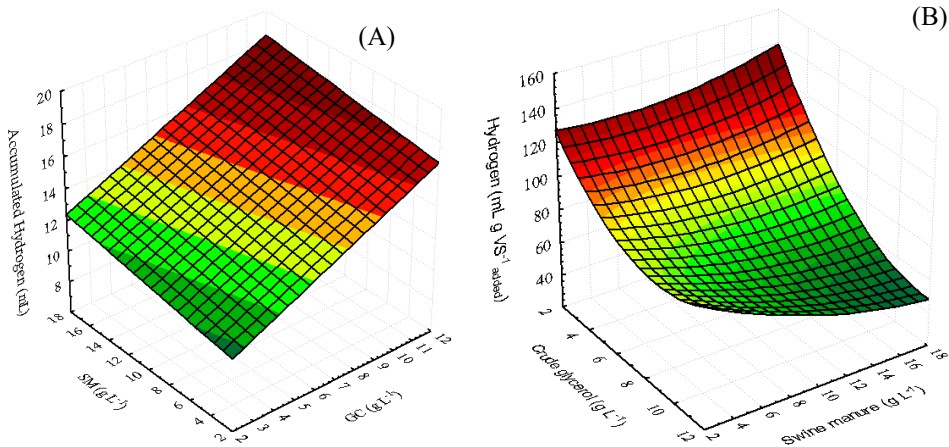

**Figure 5.** Results of accumulated hydrogen, applying the response surface methodology (RSM). (**A**) The hydrogen production yield per gram of volatile solid; (**B**) accumulated hydrogen productivity, in relation to the mixture of crude glycerol and swine manure.

The three-dimensional response surfaces displayed in Figure 5 were constructed based on Equation (1) and Equation (2) showing the influence of the variables on hydrogen production. As predicted by the model, it was possible to observe that the best hydrogen yields was 142.46 mL per gram of volatile solid added. These results were achieved when the CG factor was used, with 2.75 g L$^{-1}$ and an SM factor of 10 g L$^{-1}$. An observed fact was that the highest hydrogen production was observed with the upper-limit values (Figure 5) was designated for the SM concentration.

A high C/N ratio, as presented by the CG (1760.80 g g$^{-1}$), led to the faster consumption of carbon by microorganisms, causing a lower production of hydrogen and an increase in the concentration of free fatty acids in the environment, as observed in assays TCG100 and ACG100 with hydrogen production of 33.70 and 8.46 mL g$^{-1}$ VS added respectively. Also, a low C/N ratio (12.14), as presented by the SM, results in the accumulation of ammonia, inhibiting the growth of the fermenting microorganisms. In Figure 5A and Table 4, the maximum cumulative hydrogen yield was 17.82 mL in assay 6 (40% CG: 60% SM) over 21 d of fermentation. However, the high organic load of 11.78 g of VS (21.24 g L$^{-1}$), the high concentration of 36.35 C/N ratio and the high concentration of 11.24 g L$^{-1}$ of CG affected the hydrogen yield with 52.76 mL per gram of VS (Figure 5B). On the other hand, in Figure 5A (assay 5), the accumulated hydrogen was 14.49 mL, and the hydrogen yield was 142.46 mL g$^{-1}$ VS added (Figure 5B), using a mixture of 21.56% (2.75 g L$^{-1}$) of crude glycerol, and 78.44% (10 g L$^{-1}$) of swine manure, with a 18.06 C/N ratio. Assay 5 was the best assay, since it maintained stability in the reactor throughout the fermentation process, without the addition of a chemical agent to regulate the pH. This is because the concentration of the residues was 4.39 g of VS, low in comparison to assay 6. This result indicates that with a higher concentration of residues (>4.39 g of VS) and high CG concentration (>21.56%), it will have negative effect on the hydrogen yield. Likewise, the results showed that the best results agree with the reported results in the range of 15 to 35 C/N ratio [23,33,37].

### 2.3.4. Volatile Fatty Acids

The main metabolites (volatile fatty acids, VFAs) present in the fermentation tests were acetic acid, butyric acid, and succinic acid (Figure 6). It should be noted that the concentration of acetic acid did not differ between some fermentation assays, presenting an average concentration of 48.66 g $L^{-1}$. Bharathiraja et al. 2016 and Amaral et al. 2009 [3,38] report that in dark fermentation processes with mixed microflora, low molecular weight fatty acids are predominant; among them, the main acids present are: acetic acid, butyric acid, ethanol, *n*-butanol, acetate, and succinic acid (Figure 6).

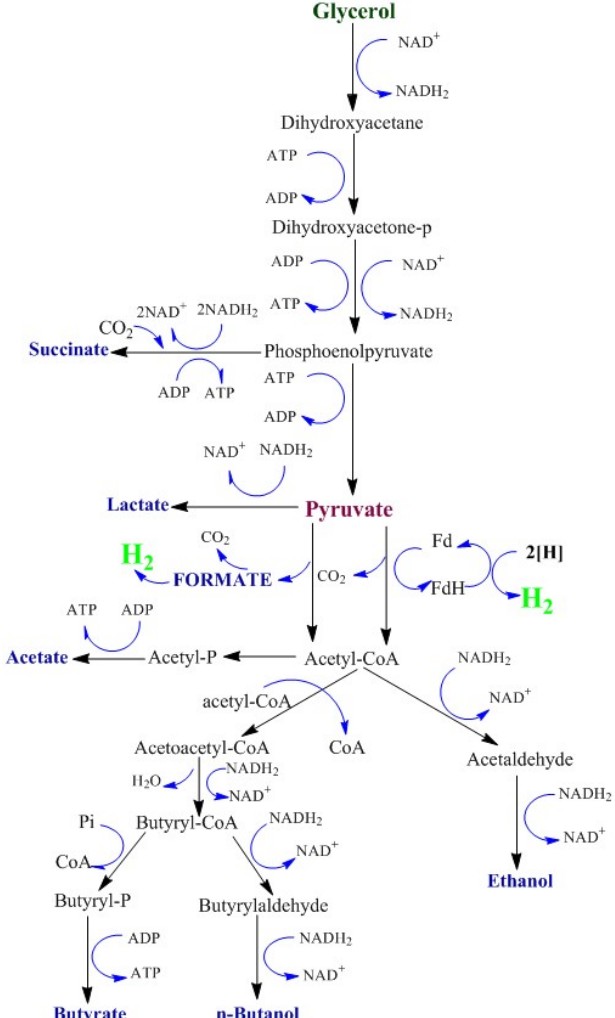

**Figure 6.** Production of hydrogen and volatile fatty acids by different metabolic routes, with different microorganisms from glycerol. Adapted by [38,39].

Furthermore, as reported by other authors the fermentative metabolism of glycerol has been studied in great detail, and it involves the participation of several bacteria such as *Citrobacter*, *Klebsiella*, *Enterobacter*, and *Clostridium* [32,35]. The assimilation of glycerol by these microorganisms is strictly linked to its ability to synthesize 1,3-propanediol (1,3-PDO) (not determined) and VFAs. The energy content of pure glycerol is 19.0 MJ $kg^{-1}$. However, for the crude glycerol of this study, it was 28.68 MJ $kg^{-1}$, possibly due to the presence of methanol and biodiesel. This high-energy content of crude glycerol indicates its potential to be an effective carbon source in the production of hydrogen, acetic acid, butyric acid, lactic acid, ethanol 1,3-PDO, etc. (Figure 6) [40,41].

The high concentration of VFAs in the fermentation process (Table 5) may be responsible for the reduction of the pH (pH 4–6), leading to a lower yield of hydrogen (Table 5). The low yields of hydrogen

caused by the accumulation of VFAs can be solved by using a second reactor with acetogenic and methanogenic bacteria, representing a two-stage system. This study provides an opportunity to value the pretreatments from anaerobic fermentation, obtaining VFAs that could be used as substrate later in the anaerobic digestion process to produce biomethane [6]. Guwy et al. 2011 [42] reported that the VFAs produced can be further valorized by integrating them into a bioelectrochemical system, such as microbial fuel cells and/or microbial electrolysis cells that are associated with the dark fermentation system. They also report that this strategy allows for better hydrogen yields, generating up to 12 moles of $H_2$ per mole of hexose.

**Table 5.** Parameters of experimental assays for producing hydrogen: Combinations of substrates (crude glycerol and swine manure), volatile fatty acids, hydrogen yields, and final pH.

| Assay | Acetic Acid (g $L^{-1}$) | Butyric Acid (g $L^{-1}$) | Propionate Acid (g $L^{-1}$) | Final pH |
|---|---|---|---|---|
| 1 | 48.69 | 0.00 | 0.00 | 5.17 |
| 2 | 48.70 | 0.00 | 0.08 | 5.17 |
| 3 | 0.00 | 0.00 | 0.00 | 5.89 |
| 4 | 48.69 | 0.00 | 0.10 | 5.92 |
| 5 | 48.68 | 0.00 | 0.07 | 5.88 |
| 6 | 48.70 | 68.15 | 0.05 | 4.77 |
| 7 | 48.69 | 0.00 | 0.00 | 5.89 |
| 8 | 48.67 | 68.08 | 0.07 | 5.14 |
| 9 | 48.67 | 0.00 | 0.03 | 4.93 |
| 10 | 48.67 | 68.15 | 0.05 | 4.72 |
| 11 | 48.67 | 68.10 | 0.03 | 5.02 |
| 12 | 48.67 | 0.00 | 0.03 | 5.03 |

## 3. Discussion

This study compared two different pretreatments (thermal and acid) on the fermentative process from crude glycerol and swine manure. We observed the influence of the pretreatment method on the hydrogen yield and the different concentrations of acetic acid, butyric acid, and propionic acid. According to the studies of dark fermentation with mixed cultures, it generates propionate, ethanol and lactic acid. Likewise, several authors have successfully used different methods of pretreatment for the inoculum to eliminate the methanogenic microorganisms, which are non-spore-forming. These treatments are mainly: thermal, typically at 100 °C for a period of 15 to 120 min, by acid pH (1–2 units) and basic pH (9–10 units) for 24 h [23,43–45]. The benefits of thermal pretreatment (105 °C, 1 atm for 60 min) and the prevalence of hydrogen-producing bacteria addressed in this research becomes appropriate, since it generates hydrogen and is economically viable, because no chemical agents are required to regulate the inoculum pH after pretreatment. The assays inoculated with thermal pretreatment achieved the highest hydrogen yield of 126.14 mL per gram of volatile solid added, with a mixture of substrates of 20% CG:80% SM (TCG20) and 16.24 C/N ratio. Also, the assays generated acetic acid, butyric acid, and lactic acid, referred to as mixed acid fermentation. However, the residues evaluated individually generated low hydrogen yield, the SM with thermal and acid pretreated inoculum generated 61.25 and 12.94 mL per gram of volatile solids respectively. Kotsopoulos in 2009 [8] used a continuously stirred tank reactor with pig slurry, under hyperthermophilic (70 °C) conditions for hydrogen production. During the steady-state period, the mean hydrogen yield was 3.65 mL g$^{-1}$ of volatile solid added, but the high level of nitrogen of pig slurry generated low rate of hydrogen. Also, the assays incubated at 70 °C make the process unviable due to the high costs in the energy consumption. In the same way, in our study, the crude glycerol with thermal and acid pretreated inocula produced a low hydrogen yield of 33.70 and 8.46 mL per gram of volatile solid added, respectively. Similarly, in the study by Akutsu in 2009 [46] used pure glycerol as a substrate in the production of hydrogen, generating a yield of 11.5 mL per gram of COD added. This low yield of hydrogen is due to the impurities and the high concentrations of carbon in the glycerol, with a C/N

ratio of 1760.80. Some authors suggest a C/N ratio of between 15–35, this value can be reached with a co-substrate, such as swine manure, since SM are residues with a high nitrogen content [47,48].

According to the results in the selection of inoculum, it was observed that the thermal pretreatment was the appropriate pretreatment for producing hydrogen from dark fermentation. Likewise, this study evaluated the increase in the concentration of the mixture of CG and SM, to optimize the process of hydrogen production. For this, a design of experiments was carried out using a range of 4 to 10 g L$^{-1}$ for CG and SM in a range of 5 to 15 g L$^{-1}$. According to the results, it was observed that the crude glycerol and swine manure factors were significant in the linear analysis, with *p*-values of 0.003 and 0.01, with a level of significance of 5%. In the same way, the quadratic effect of the crude glycerol factor was positive, with a *p*-value <0.5. The SM factor in the quadratic effect was not significant with *p*-value of 0.69. However, this indicates that increasing the concentration of glycerol as a substrate positively influences the response in the production of hydrogen. The increase in the concentration of CG in the fermentative process can have a positive effect on hydrogen production, but within a certain proportion. The best assay was assay 5, with a cumulative hydrogen production of 14.49 mL and 142.46 mL per gram of VS added to the reactor, after 21 days of fermentation. These results were obtained by the addition of 10 g L$^{-1}$ of SM and 2.75 g L$^{-1}$ of CG, giving a combined C/N ratio of 18.06. If the CG concentration is above 21.56% of the mixture of residues, it is likely that the hydrogen yield is affected. This effect is used because the carbohydrates present in the CG are consumed faster by the fermenting microorganism, increasing the concentration of VFAs in the system; as a consequence, a pH lower than 5.5 was reached. The optimum pH for hydrogen production should be between 5.5 and 6.5. Therefore, this study suggests that the concentration of the mixture of substrates should be 12.75 g L$^{-1}$, with 21.56% CG and 78.44% SM, to produce hydrogen successfully without the addition of chemical agents to regulate the pH. It should be noted that these results are of interest for generating hydrogen on a larger scale. Thus, the experimental results and the surface response methodology (RSM) carried out in this study was able to determine the location of the optimal values of crude glycerol and swine manure that maximized the yield of hydrogen in the fermentative process. Statistical approaches such as RSM were employed to maximize the hydrogen production of CG/SM ratios, by optimizing the factors with few assays and less operational cost.

## 4. Materials and Methods

### 4.1. Feedstock and Enrichment of Inoculum

The mixed anaerobic microflora was acquired from an anaerobic methane production reactor in operation at the Universidade Federal dos Vales do Jequitinhonha e Mucuri (UFVJM-Campus JK, Diamantina MG, Brazil) to be used as the inoculum. The sewage was pretreated by two methods, to selectively enrich the hydrogen-producing mixed microflora, inhibiting the growth of methanogenic bacteria and facilitating the growth of sporulating bacteria. The pretreatment methods were: thermal (105 °C, 1 atm for 60 min) and acid (1 mol L$^{-1}$ HCl, pH 2 for 24 hr at constant homogenization). After the acid treatment, the pH of the inoculum was neutralized (pH = 7).

All of the reagents and solvents used in this work were analytical grade, P.A.-A.C.S (Sigma-Aldrich, St. Louis, MO, USA and Honeywell, Charlotte, CN, USA). The analytical standards used for liquid chromatography had a minimum purity of 99% (Sigma-Aldrich, St. Louis, MO, USA). The CG was a byproduct of biodiesel production from research carried out in the postgraduate program on biofuels at UFVJM (Diamantina MG, Brazil). The SM was obtained from the swine breeding of the Zootechnical Program of the same University.

### 4.2. Physicochemical Characterization of Residues

The physical and chemical characterization of the substrates included chemical oxygen demand (COD), total solids (TS), volatile solids (VS), and pH, which were determined according to the method of the American Public Health Association (1995) [49]. All analyzes were done in triplicate.

The percentage of carbon, nitrogen, and hydrogen in crude glycerol, and swine manure were determined by elemental analysis (Element Analyzer CHNS/O TruSpec® Micro, LECO, MI, USA).

### 4.3. Comparison of Hydrogen-Producing Bacteria

The experiments were designed to evaluate the influence and interaction of CG and SM on hydrogen production and VFAs. All experiments were run in batch mode, using bottles of 50 mL volume. The evaluation of two different pretreatment methods (thermal and acid) was performed to determine the best pretreatment for the selection of hydrogen-producing bacteria. Reactors were inoculated with 5 g VS of pretreated anaerobic inoculum, and fed with different substrate concentrations, according to Table 6, under anaerobic conditions at 30 °C. At the same time, two assays with inoculates (thermal and acid) without substrates (CG or SM) were incubated as a control experiment.

**Table 6.** Experimental essays of hydrogen yield from CG and SM with two different inocula.

| Assay with Inoculum Pretreated with Acid | Assay with Thermally Pretreated Inoculum | CG (%) | SM (%) | COD (g L$^{-1}$) |
|---|---|---|---|---|
| ACG100 | TCG100 | 0 | 100 | 4.59 ± 0.18 |
| ASM100 | TSM100 | 100 | 0 | 4.54 ± 0.18 |
| ACG80 | TCG80 | 80 | 20 | 4.49 ± 0.17 |
| ACG60 | TCG60 | 60 | 40 | 4.45 ± 0.17 |
| ACG40 | TCG40 | 40 | 60 | 4.65 ± 0.18 |
| ACG20 | TCG20 | 20 | 80 | 3.20 ± 0.12 |

COD: chemical oxygen demand.

### 4.4. Experimental Planning for Hydrogen Production

The assays for the anaerobic fermentative study were delimited by an experimental design that used a central composite design, as presented in Table 7, containing two factors; four axial points and four central points. The factors evaluated were the concentrations of CG and of SM.

**Table 7.** Matrix of the central composite design containing four axial points and four central points, used to study the hydrogen production of CG with SM.

| Factors | Axial ($-\alpha$) | Minimum ($-1$) | Central Point (0) | Maximum (+1) | Axial ($+\alpha$) |
|---|---|---|---|---|---|
| Crude Glycerol (g L$^{-1}$) | 2.75 | 4.00 | 7.00 | 10.00 | 11.24 |
| Swine Manure (g L$^{-1}$) | 2.93 | 5.00 | 10.00 | 15.00 | 17.07 |

The assays were carried out to evaluate the influence and interaction of the mixture of crude glycerol and swine manure on the hydrogen yield. All of the experiments were carried out in batch mode, using glass bottles with a capacity of 50 mL. The reactors were inoculated with 5 g of VS of pretreated sludge (by thermal pretreatment). After addition of the pretreated sludge, the reactors received different concentrations of CG and SM substrates. The assays had inoculum/substrate (I/S) ratios in a range of 0.44 to 1.16 (Table 8).

The nutrient solution used was an adaptation of the one used by Chernicharo 2007 [50], Aguilar-Aguilar et al., 2017 [9], and Aquino et al., 2007 [51] (Table 9). The pH of the aqueous phase was determined by mixing the crude glycerol and swine manure. After the loading of the inoculum, nutrients, and substrates, the bottles were firmly sealed with a rubber septum (butyl rubber) to maintain an anaerobic environment. All experiments were kept in a water bath heated at 30 °C. The anaerobic fermentation was finished until no gas was produced.

**Table 8.** Concentrations of CG and SM assigned by the central composite design, used as planning for the anaerobic assays.

| Assays | CG (g L$^{-1}$) | SM (g L$^{-1}$) | (I/S) Ratio (g VS g VS$^{-1}$) |
|--------|--------|--------|--------|
| 1 | 4.00 | 5.00 | 1.16 |
| 2 | 4.00 | 15.00 | 0.83 |
| 3 | 10.00 | 5.00 | 0.52 |
| 4 | 10.00 | 15.00 | 0.45 |
| 5 | 2.75 | 10.00 | 1.23 |
| 6 | 11.24 | 10.00 | 0.44 |
| 7 | 7.00 | 2.93 | 0.76 |
| 8 | 7.00 | 17.07 | 0.56 |
| 9 | 7.00 | 10.00 | 0.64 |
| 10 | 7.00 | 10.00 | 0.64 |
| 11 | 7.00 | 10.00 | 0.64 |
| 12 | 7.00 | 10.00 | 0.64 |

CG: crude glycerol, SM: swine manure, COD: chemical oxygen demand, VS: volatile solids.

**Table 9.** The nutrient solution, micro- and macro-nutrients (adapted from Chernicharo 2007 [50], Aguilar-Aguilar et al., 2017 [9], Aquino et al., 2007 [51]).

| Macro Nutrients | Concentration (mg L$^{-1}$) | Micro Nutrients | Concentration (mg L$^{-1}$) |
|--------|--------|--------|--------|
| $NH_4Cl$ | 1112 | $FeCl_3 \cdot 6H_2O$ | 5 |
| $(NH_4)H_2PO_4$ | 132.5 | $ZnCl_2$ | 0.13 |
| $(NH_4)_2HPO_4$ | 44.5 | $MnCl_2 \cdot 4H_2O$ | 1.25 |
| $MgCl_2 \cdot 6H_2O$ | 250 | $(NH_4)_6MO_7O_{24} \cdot 4H_2O$ | 1.6 |
| $CaCl_2 \cdot 2H_2O$ | 189 | $AlCl_3 \cdot 6H_2O$ | 0.13 |
| $NaHCO_3$ | 2500 | $CoCl_2 \cdot 6H_2O$ | 5 |
| - | - | $NICl_2 \cdot 6H_2O$ | 13 |
| - | - | $H_3BO_3$ | 3 |
| - | - | $CuCl_2 \cdot 2H_2O$ | 8 |
| - | - | HCl | 1 |

The dependent variables or response factors of choice for the experimental analysis were the normalized volumes of hydrogen yield (mL). Statistica 7.0 software was used for the analysis of the hydrogen response, to allow the description of the results by the means of the adjustment to a linear mathematical model (Equation (3)) or a quadratic model (Equation (4)), used for the generation of response surface curves, and the determination of possible interactions [9,32]. Data were submitted for the analysis of variance, considering a level of significance ($\alpha$) of 0.05. The coefficient of determination ($R^2$adj) was used as a parameter of adequacy of the mathematical models generated by a least squares regression of the phenomena evaluated.

$$Y = \beta_0 + \beta_1 x_1 + \beta_2 x_2 + \beta_{12} x_1 x_2 + \varepsilon \tag{3}$$

where:

$Y$ = value of the dependent variable;

$x_1$ e $x_2$ = independents variables

$\beta_0, \beta_1, \beta_2, \beta_{12}$ = parameters of the regression model for each variable;

$\varepsilon$ = error term (effects not explained by the model).

$$Y = \beta_0 + \sum_{i=1}^{k} \beta_i x_i + \sum_{i=1}^{k} \beta_{ii} x_i^2 + \sum \sum_{i<j} \beta_{ij} x_i x_i + \varepsilon \tag{4}$$

where:

$Y$ = value of the dependent variable;

$x_1$ e $x_2$ = independents variables;

$\beta_0$, $\beta_i$, $\beta_{ii}$, $\beta_{ij}$ = parameters of the regression model for each variable;

$\varepsilon$ = error term (effects not explained by the model).

The variance analysis was used to identify the factor that had the greatest influence on the response, and to examine the adequacy of the model. To do this, the level of significance of the estimated coefficients and the model fit was tested with the *p*-statistic.

### 4.5. Quantification of the Volume of Hydrogen

The volume of hydrogen produced in each reactor was measured with an inverted glass bottle system (Figure 7). The hydrogen produced passes directly to the inverted flask containing a 3N NaOH (120 g L$^{-1}$) solution, where the biogas is bubbled to eliminate the $CO_2$, leaving only hydrogen in the upper part of the bottle (II). With the help of a test tube, the volume displacement of water was measured, which was directly proportional to the volume of gas that was produced in each reactor (III). Hydrogen values are expressed at normal conditions of temperature and pressure (NTP, 0 °C and 1 atm).

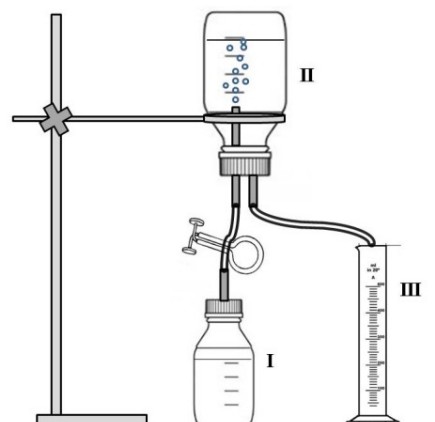

**Figure 7.** Volume determination apparatus for hydrogen produced. (I) Bioreactor with a biogas outlet, (II) inverted flask with a 3 N NaOH solution, and (III) test tube for the quantification of bioH$_2$ volume (Aguilar-Aguilar et al., 2017 [9]).

### 4.6. Quantification of Glycerol and Organic Acids

The VFAs (volatile fatty acids) were quantified at 220 nm by liquid chromatography, with the use of a Shimadzu Prominence UFLC 20A equipped with a Rezex ROA-Shodex$^{\text{TM}}$ column (300 × 7.5 mm), with a UV/Vis detector at 60 °C. injections of 5 µL volumes were done by using a programmed automatic injector. A 0.0025 M $H_2SO_4$ solution at 0.6 mL min$^{-1}$ was used as a mobile phase. Compound identification was done using external standards.

### 5. Conclusions

In this study, it was demonstrated that the use of inoculum that was enriched with acid and thermal pretreatment on the fermentative process of crude glycerol and swine manure influenced the hydrogen yield. However, the use of thermal pretreatment was better, since it needed less pretreatment time, a lower cost of chemical reagents for the treatment of inoculum, and a higher hydrogen yield. The application of the experimental design tool effectively contributed to the modeling and prediction of the best relationship between the crude glycerol and the swine manure in the fermentation process, as confirmed by the high levels of significance determined from the central composite planning. The assays inoculated with thermal pretreatment achieved the highest hydrogen yields of 142.46 mL per gram of volatile solid, using 12.75 g of mixture of substrates (21.56% CG:78.43% SM) and an 18.06 C/N ratio. We were able to find a specific concentration of the residual mixture that maximized the hydrogen yield with few experimental assays. These results demonstrate that the swine manure

and crude glycerol are ideal substrates in the fermentation process, with a synergy between the mixture of residues and the inoculum with thermal pretreatment.

**Author Contributions:** Conceptualization, A.-A.F.A.; Methodology, A.-A.F.A.; Investigation, A.-A.F.A.; Writing-Original Draft Preparation, A.-A.F.A.; Writing-Review & Editing, A.-A.F.A., A.L., J.A.U. and P.J.S.; Visualization, A.-A.F.A., A.L.; Supervision, P.J.S.; Project Administration, P.J.S.; Funding Acquisition, P.J.S.; A.-A.F.A.

**Funding:** The financial support for this work was given through the projects DGAPA-UNAM IN109319 and PE210918.

**Acknowledgments:** The authors acknowledge the support of the postgraduate program in biofuels of the Universidade Federal dos Vales do Jequitinhonha e Mucuri (UFVJM). Aguilar-Aguilar F.A. acknowledges the scholarship (No. 402762) given by CONACYT, Mexico.

**Conflicts of Interest:** The authors declare that they do not have any conflict of interest.

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
