# Peer review of "Optimization of Hydrogen Yield from the Anaerobic Digestion of Crude Glycerol and Swine Manure"

_catalysts, doi:10.3390/catal9040316_

Round 1
Reviewer 1 Report
This paper deals with the optimization of hydrogen evolution from mixtures of crude glycerol (CG) and swine mature (SM) by response surface methodology (RSM). The RSM results (Figure 5) shows that the condition of about 7 g L-1 of CG and about 17 g L-1 of SM provided the largest amount of evolved hydrogen (about 17 mL). However, it is explained in many parts that the maximum hydrogen yield was obtained by the addition of 2.75 g L-1 of CG and 10 g L-1 of SM. The RSM results shown in Figure 5 are inconsistent with the description of the RSM results. Moreover, it is explained in the end of the introduction section that the aim of this research was to explore the viability of SM by co-digestion with CG for batch experiments of hydrogen production under mesophilic condition. However, in the abstract and conclusion sections, only the RSM results were described. I wonder what the purpose of this study is. Furthermore, it is explained in the Materials and Methods section that the experiments were designed to evaluate the influence of the interaction of CG and SM on hydrogen production. However, the interaction of CG and SM was not described, based on the RSM results. It is explained on the bottom of page 4 that this study demonstrated that the high concentration of CG increased the C/N ratio and decreased the hydrogen yield. However, this explanation is inconsistent with the RSM results shown in Figure 5. Thus, in many parts of this paper, incoherent descriptions are provided. It is hard for me to follow what is described and what the novelty of this study is. So, I do not feel that this paper reach the level suitable for publication. Other comments are shown below:
1. If the condition providing the highest hydrogen yield per volatile solids is different from the one providing the largest amount of hydrogen, RSM analysis should also be conducted for hydrogen yield per volatile solids.
2. The C/N ratios of CG and MS shown in Table 1 are different from the C/N ratios of 100% CG and 100% MS samples shown in Table 2. Why?
3. What is “EP” in Table 3?
4. What is the predicted values in Figure 3? Are they the values calculated using Equation 1?
5. Two X 2 terms are present in Equation 1.
6. “Equation 1” and “Equation 2” on page 15 should be replaced with “Equation 2” and “Equation 3”.
7. The English is not so good.
Author Response
1) The manuscript was corrected according to the observations of the reviewer, which can be seen from line 291 to 307. In figure 5A and table 4 the maximum cumulative hydrogen yield is indicated as 17.82 mL for the assay 6 (40 % CG: 60 % SM) in 21 days of fermentation. However, the high organic load of 11. 78 g of VS (21.24 g L-1), high concentration (36.35) of C/N ratio and the high concentration (11.24 g L-1) of CG affected the hydrogen yield with 52.76 mL per gram of VS (Figure 5B). On the other hand, in figure 5A for the assay 5 the maximum yield of accumulated hydrogen was 14.49 mL and the hydrogen yield was 142. 46 mL g-1 of VS added (Figure 5B), using the mixture of 21.56 % (2.75g L-1) of crude glycerol and 78.44 % (10 g L-1) of swine manure and 18.06 C/N ratio. The assay 5 was the best assay, since it maintained pH stability in the reactor throughout the fermentation, without the addition of a chemical agent to regulate the pH. This fact is due to the low concentration of residues, 4.39 g of VS, in comparison to assay 6. This result indicates that a higher concentration of residues (> 4.39 g of VS) and high CG concentration (21.56 %) will have negative effect on the hydrogen yield. The other treatments did not show a significant increase in hydrogen yields. These details are included in the revised manuscript.
2) The crude glycerol has a C/N ratio of 1760.80 and swine manure has a C/N ratio of 12.14. In table 1 the C/N ratio of individual substrates are presented. In table 2 it is presented the C/N ratio of the mixtures. The data is correct.
3) The Spanish term in the manuscript was corrected according to the observations of the reviewer.
4) As mentioned in the materials and methods from line 449 to 453, Statistica 7.0 software was used for the analysis of the hydrogen response to allow the description of the results by means of the adjustment to a linear mathematical model (Equation 3) or a quadratic model (Equation 4) used for the generation of response surface curves and the determination of possible interactions. In the analysis of the response surface graph generated from Equation 1 (Figure 5A) and Equation 2 (Figure 5B), it was possible to observe the effect of CG and SM factors on hydrogen production (Line 276 to 278). The adjustment of the polynomial model generated for the production of hydrogen as a function of different ratios of CG and SM yielded a determination coefficient (R2adj). The model generated a positive response from the experimental data (hydrogen yield) ( line 256 to 261).
5) The equations were corrected according to the observations of the reviewer.
6) The numbers of the equations were corrected.
7) The entire manuscript was thoroughly revised, and English corrections were done.
Reviewer 2 Report
The manuscript deals with the anaerobic digestion of crude glycerol and swine manure as hydrogen production method. The results are clearly stated in the manuscript, but there is a minor remark to the text of the manuscript:
1). In «Abstract» the abbreviations (CG, SM, RSM, C/N) should not be used.
2). The numbers should be separated from the unit by a space (for example: in Abstract – 21.56 %, 78.43 %). Check all the text!!!;
3). CAS should be given for each of the compounds used in the experiments (in Section «Materials and methods»).
4). What is the experimental error of kinetic measurements?
5). In section "Materials and Methods" the bacterial strains used in this study should be indicated.
6). What are the reasons of decrease in gas generation rate during the experiment? In section «Discussion» the inhibition problem should be discussed.
In my opinion, the interesting problem is discussed but the manuscript must be send to other Journal - Applied Sciences, Energies, Fermentation, Resources.
Best regards, Reviewer.
Author Response
The manuscript was corrected according to the observations of the reviewer.
The manuscript was corrected according to the observations of the reviewer.
All the reagents and solvents used in this work were analytical grade, P.A.-A.C.S. The analytical standards used for liquid chromatography had a minimum purity of 99%.
The values of biogas production are represented by the mean of volume of biogas at the end of the kinetic for each gram of volatile solids (Table 4). However, for a better presentation of the results, the mean of each assay was plotted without the standard error. All experiments were done in triplicate.
In materials and methods, line 397 to 403 it is mentioned that the mixed anaerobic microflora acquired from an anaerobic methane production reactor in operation. Subsequently, the sewage sludge was pretreated by two methods to selectively enrich the hydrogen-producing mixed microflora, inhibiting the growth of methanogenic bacteria and facilitating the growth of sporulation bacteria. As reported by other authors the fermentative metabolism of glycerol has been studied in detail and involves participation of several bacteria such as Citrobacter, Klebsiella, Enterobacter and Clostridium (Line 318 to 320).
As mentioned in line 377 to 389, the hydrogen production depends on the C/N ratio, if a substrate is abundant in carbohydrates, such as crude glycerol, it increases the risk of generating low hydrogen yield. If the substrate is rich in nitrogen, such as swine manure, there is a probability of inhibiting the hydrogen-producing bacteria with increased ammonium and ammonia concentration in the reactor. In this way, we propose that the best waste mixture was 12.75 g L-1 (21.56% CG and 78.44% SM) to maintain a pH of 5.88 and successful hydrogen production. One can also observe the interaction of waste with the decrease of biogas as mentioned in lines 127 to 137 and 203 to 214.
We received the invitation to submit the article to the Special Issue "New Glycerol Upgrading Processes". We consider that this research achieves the objective of the special issue, since it provides scientific interest data in bioprocesses. In this study, we verified that the crude glycerol combined with swine manure is an adequate mixture of substrate to generate hydrogen at low cost. Likewise, we demonstrate that combining a bioprocess with the response surface methodology can generate enough information to maximize the production of hydrogen, with few assays, in a short time and with less cost. The information generated from this study can serve as the basis for further studies of large-scale hydrogen production.
Reviewer 3 Report
Suggestions to the authors:
The authors carried out an interesting work for obtaining hydrogen from different sources and the results seem to be promised for future research in the field of renewable energies. Then I quote a few suggestions to improve the current version of this work.
1.- The first sentence in abstract must be redacted again “This reason this study”
2.- Revise reference “Kotsopoulos 2009 [9]” in section introduction and paragraph 51st. In the same line, revise in section 2.2. Inoculum selection with thermal and acid pretreatments in hydrogen production and paragraph 155th “this estudy”.
3.- Figures 1 and 2 must be revised because present an ungraceful format for a scientific journal. Figure 5 has poor format , for example axes are not clarity.
4.- Which software was used to carry out the response surface methodology ? It must be included in manuscript.
Author Response
Comments 1 and 2. The entire manuscript was thoroughly revised, and English corrections were done.
All figures were modified to improve their quality.
Statistica 7.0 software was used for the analysis of the hydrogen response to allow the description of the results by means of the adjustment to a linear mathematical model (Equation 3) or a quadratic model (Equation 4) used for the generation of response surface curves and the determination of possible interactions (Line 449 to 456). It is mentioned in the manuscript.
Round 2
Reviewer 1 Report
I think that the revision improved this paper. However, before publication, careless mistakes should be corrected.
1. In the abstract, it is explained, “The maximum hydrogen yield was 14.49 mL (accumulated) and 142.46 mL per gram of volatile solid added.” The maximum hydrogen yield per gram of volatile solid added was 142.46 mL for assay 5; however, the maximum accumulated hydrogen yield was not 14.49 mL. I suggest that “14.49 mL (accumulated) and” should be omitted.
2. The C/N ratio of 100% SM is 12.14 in Table 1 but 1760.80 in Table 2. The C/N ratio of 100% CG is 1760.80 in Table 1 but 12.14 in Table 2.
3. The footnote of Table 5 should be omitted.
4. In line 372 (page 12), it is explained, “The best assay was the assay 4…” I think that “the assay 4” should be replaced with “the assay 5”.
5. In Table 8, the sample names of 100% SM are A100CG and T100CG, and the ones of 100% CG are A100SM and T100SM. Are they right? Moreover, the sample names “TCG100”, “TSM100”, “ACG100”, and “ASM100” are used in Figure 1; however, these are different from the ones shown in Table 8.
Author Response
According to reviewer's suggestions it was modified line 21 to line 25. “The maximum hydrogen yield was 142.46 mL per gram of volatile solid added. It was used 21.56 % of crude glycerol (2.75 g L-1) and 78.44 % (10 g L-1) of swine manure, maintaining a carbon/nitrogen ratio of 18.06, with a fermentation time of 21 days.”
The errors in Tables 1 and 2 were corrected.
The footnote of Table 5 was omitted as suggested by the reviewer.
The error was corrected, replacing the assay 4 for assay 5.
The nomenclature of the assays was corrected in table 8 and figure 1. In the manuscript the nomenclature A100CG and T100CG were replaced by ACG100 and TCG100 respectively.
Reviewer 2 Report
I consider that the paper has been improved according to Reviewer’s recommendations. I recommend this paper to be accepted for the publication in Journal «Catalysts».
Best regards, Reviewer
Author Response
We believe that the design of the research is adequate for the study of hydrogen production from the different combinations of crude glycerol and swine manure. The aim of the experimental design suggests several advantages in comparison to the conventional optimization methods. It helps to the compression of the variables (CG/SM) and the response (hydrogen), improves the yields of the process reducing variability and time of development, as well as experimental costs. Statistical methods such as the response surface methodology have already been used successfully for optimization of several processes and bioprocesses.